# Design and Analysis of Brake-by-Wire Unit Based on Direct Drive Pump–Valve Cooperative

Peng Yu [1,2], Zhaoyue Sun [2,*], Haoli Xu [2], Yunyun Ren [3] and Cao Tan [1,2,*]

1   State Key Laboratory of Automotive Simulation and Control, Jilin University, Changchun 130022, China; yupeng199910@163.com
2   School of Transportation and Vehicle Engineering, Shandong University of Technology, Zibo 255000, China; utxuhl@yeah.net
3   Shuntai Automobile Co., Ltd., Zibo 255000, China; renyy123123@163.com
*   Correspondence: chsunzy@outlook.com (Z.S.); njusttancao@yeah.net (C.T.);
    Tel.: +86-13053308225 (Z.S.); +86-0533-2786837 (C.T.)

**Abstract:** Aiming at the requirements of distributed braking and advanced automatic driving, a brake-by-wire unit based on a direct drive pump–valve cooperative is proposed. To realize the wheel cylinder pressure regulation, the hydraulic pump is directly driven by the electromagnetic linear actuator coordinates with the active valve. It has the advantages of rapid response and no deterioration of wheel side space and unsprung mass. Firstly, by analyzing the working characteristics and braking performance requirements of the braking unit, the key parameters of the system are matched. Then, in order to ensure the accuracy of the simulation model, the co-simulation model of the brake unit is established based on the Simulink-AMESim co-simulation platform. Then, the influence law of key parameters on the control performance is analyzed. Finally, the experimental platform of the brake unit is established. The accuracy of the co-simulation model and the feasibility of the brake-by-wire unit based on direct drive pump–valve cooperative are verified through the pressure control experiment and ABS simulation, which shows that the braking unit has good dynamic response and steady-state tracking effect.

**Keywords:** electric vehicle; brake-by-wire; electro-hydraulic co-simulation; parameter matching; characteristic analysis

## 1. Introduction

With the development of the vehicle chassis toward the direction of X-by-wire chassis and slide chassis, the demand for distributed braking systems is becoming more and more intense. At the same time, high-level automatic driving puts forward higher requirements for the response speed and control accuracy of the drive-by-wire system [1–4].

The Electronic Hydraulic Brake (EHB) System has been widely favored due to its advantages, such as rapid response, high power density, and brake structure compatibility [5–9]. In 2013, Bosch launched the I-Booster, a motor servo booster that is independent of vacuum booster, which is a typical motor servo EHB system. TRW introduced a high-pressure accumulator-based EHB system called the SCB, which has a brake master cylinder consisting of front and rear chambers and parallel pistons. Since the pistons can be moved fore and aft and the front part is connected to the front wheel, the pressure of the system can be adjusted through the rear part of the pistons [10]. In 2010, Hitachi launched the EHB system named e-ACT, which uses an electric motor to drive an actuator that pushes a master cylinder piston, and the rotational force of the motor is converted into linear motion by a ball screw [11]. In 2021, Continental launched a new generation of EHB system MKC2 based on MKC1, which uses a multi-logic framework with independent partitions, successfully reducing the number of brake system components, and its modular and scalable design provides ideas for future dynamic control of brake systems.

At present, the EHB system is still the mainstream of the development of brake-by-wire systems. In order to take into account the needs of the market and efficiency, it is of great significance to explore a new brake-by-wire structure to improve the performance of brake-by-wire and simplify the system structure [12–16]. Sun designed and developed an integrated master cylinder and decoupling electro-hydraulic composite braking system and proposed the hydraulic braking force distribution strategy [17]. Wu designed an integrated EHB system. By controlling the solenoid valve and designing the working mode switching control strategy, the working mode switching was realized to achieve high redundancy and independent control of the four-wheel cylinders of the braking system [18]. EHB with different characteristics has been proposed to promote the development of brake-by-wire technology, but it is difficult to directly apply to distributed braking systems [19–23]. It is necessary to design a distributed brake-by-wire unit to meet the needs of intelligent driving and meet the requirements of new energy vehicles for precise control of braking force and lightweight. Fu designed a new type of electromechanical brake with an automatic wear adjustment function and adopted a worm gear mechanism for power transmission, which has the advantage of low delay and small size [24]. Yu designed a distributed electro-hydraulic brake-by-wire system, which uses a motion conversion mechanism to change the motion of the rotating motor into linear motion and then drive the hydraulic wheel cylinder [25]. Chang designed an EHB unit that uses a linear motor to drive the hydraulic piston, amplifying the driving force of the motor through unequal-diameter hydraulic pistons and wheel cylinder pistons [26]. The existing EHBs are often improved based on proven hydraulic braking systems, which still retain complex hydraulic pipelines [27–30]. Distributed brake-by-wire systems often require the entire brake unit to be integrated into the caliper, deteriorating wheel side space and unsprung mass. Therefore, this paper designed a brake-by-wire unit based on direct drive pump-valve cooperative. The electro-magnetic linear actuator (EMLA) is directly driven by the hydraulic pump and the active valve coordination to achieve wheel cylinder pressure regulation.

The main contributions are summarized as follows: (1) A brake-by-wire unit based on direct drive pump–valve cooperative is proposed, which has the advantages of rapid response and no deterioration of wheel side space and unsprung mass; (2) Based on the Simulink-AMESim co-simulation platform, a co-simulation model of the brake unit is established and verified; (3) The characteristics of the key parameters are analyzed, and the influence law of each parameter on the control performance is investigated. The rest of this work is organized as follows. Parameter matching design of the key parameters is carried out in Section 3. The dynamic model of the brake-by-wire unit based on direct drive pump–valve cooperative is established in Section 4. The feasibility is verified by pressure experiments and ABS simulations in Section 5. Finally, a conclusion is made in Section 6.

## 2. System Scheme and Principle

In response to the requirements of distributed braking and high-level automatic driving, this paper designed a brake-by-wire unit based on a direct drive pump–valve cooperative. The structural diagram is shown in Figure 1, which mainly includes an EMLA, hydraulic pump, active valve group, brake wheel cylinder based on the existing structure, and replenishment oil tank. The EMLA, hydraulic pump, active valve group, and replenishment oil tank are integrated and installed on the vehicle frame. The oil pipe is connected to the brake wheel cylinder based on the existing structure, which effectively simplifies the brake pipe and realizes distributed braking without deteriorating the wheel side space and unsprung mass. The EMLA directly drives the reciprocating motion of the piston of the hydraulic pump, combined with the switch control of the active valve group, to realize the rapid adjustment of the driving wheel cylinder pressure. The active valve group is composed of two solenoid valves and the oil refill valve is a normally closed valve, which controls the continuity of the oil circuit between the replenishment oil tank and the direct drive pump, and the pressure maintaining valve is a normally open valve that controls the oil circuit from the pump to the brake wheel cylinder. The diameter of the

piston in the direct drive pump is smaller than the diameter of the wheel cylinder piston, and the driving force of the actuator is amplified by the principle of unequal diameter hydraulic amplification, thus obtaining sufficient braking force using a smaller actuator. As the power source of the brake unit, the EMLA completes the conversion of electrical energy to mechanical energy. The detailed working principle of using high-power density moving coil electromagnetic linear actuators is shown in references [31,32].

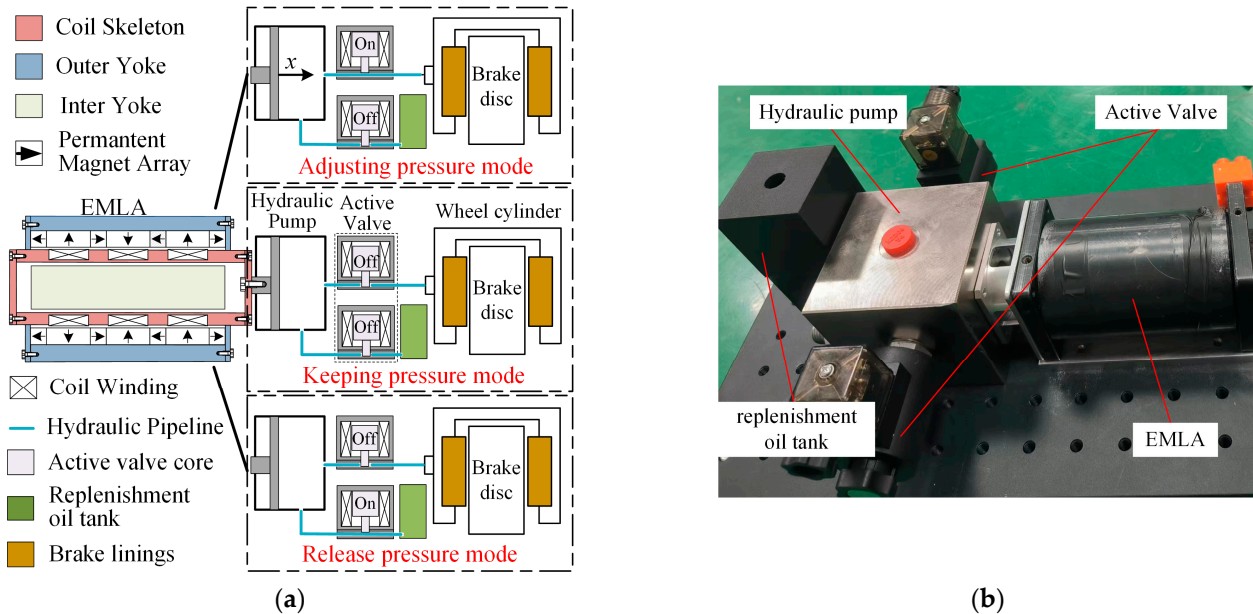

**Figure 1.** The structure of brake-by-wire unit based on direct drive pump–valve cooperative: (**a**) The schematic diagram; (**b**) The prototype.

The structure of brake-by-wire unit based on direct drive pump–valve cooperative is shown in Figure 1. The typical working process is divided into adjusting pressure mode, keeping pressure mode, and releasing pressure mode. In the releasing pressure mode, the pressure maintaining valve is opened, and the oil refill valve is closed. Under the action of brake wheel cylinder pressure and brake wheel cylinder return spring, the brake wheel cylinder hydraulic flow back hydraulic pump, actuator, and hydraulic pump piston return to the initial position. If the linear actuator does not return to the initial position due to hydraulic oil leakage or other reasons, the pressure maintaining valve is closed, the oil refill valve is opened, and the actuator drives the hydraulic pump piston back to the initial position. In the adjusting pressure mode, the pressure maintaining valve is opened, and the oil refill valve is closed. The EMLA drives the hydraulic piston, quickly discharging hydraulic oil from the hydraulic pump head into the brake wheel cylinder, pushing the brake wheel cylinder piston to eliminate brake clearance, and then quickly adjusting the brake wheel cylinder pressure by controlling the output force of the linear actuator. In the keeping pressure mode, the pressure maintaining valve and oil refill valve are closed, the actuator does not work, and brake wheel cylinder pressure is maintained. The brake unit relieves the working burden of the EMLA and reduces the working energy consumption under the keeping pressure mode.

The advantages of the brake-by-wire unit based on direct drive pump–valve cooperative include the following: (1) the realization of distributed braking while simplifying the hydraulic line and not deteriorating the unsprung mass; (2) the use of the line actuator to directly drive the piston, instead of choosing the form of motor and motion conversion mechanism, to improve the system response speed; (3) the direct drive pump does not work in the keeping pressure mode, effectively reducing the working energy consumption.

### 3. Parameter Matching Design

The key components of a brake-by-wire unit based on direct drive pump–valve cooperative are the actuator, direct drive pump, and active valve, which have a great influence on the braking performance. The following takes an X-by-wire chassis as an example to match key parameters, and X-by-wire chassis parameters are shown in Table 1.

**Table 1.** X-by-wire chassis parameters.

| Item | Value | Unit |
|---|---|---|
| Mass at full load | 300 | kg |
| Wheel radius | 228 | mm |
| Brake type | Disc brake | |
| Cross-sectional radius of the piston | 19 | mm |
| Effective radius of friction plate | 110 | mm |
| Friction coefficient of friction plate | 0.38 | |

To ensure that the electromagnetic force of the EMLA can be effectively amplified, the area of the plunger in the pump should be small to ensure sufficient amplification. However, if the plunger area is too small, it will result in too much pushrod travel and increase its axial size. Considering the magnification and axial size, the plunger area $S_1$ is set to 28 mm$^2$ so that the theoretical magnification $\varepsilon$ is

$$\varepsilon = \frac{S_2}{S_1} \tag{1}$$

where $S_2$ is the cross-sectional area of the piston. The brake unit adopts a disc brake, and the braking torque comes from the caliper clamping force; the caliper clamping force $F_c$ is

$$2F_c r_{c1} f = f_{b1} r_b \tag{2}$$

where $r_{c1}$ is the effective radius of brake lining; $f$ is the friction coefficient of brake lining; $f_{b1}$ is the maximum braking force of a single wheel, and $r_b$ is the wheel radius. The relationship between the caliper clamping force and the maximum pressure of the brake wheel cylinder is

$$F_c < P_{\max} S_2 \tag{3}$$

where $P_{\max}$ is the maximum pressure of brake wheel cylinder, calculated as 4.5 MPa. The maximum thrust of the actuator $F_{\max}$ is

$$F_{\max} = P_{\max} S_1 + F_p \tag{4}$$

where $F_p$ is the pre-tightening force of the wheel cylinder piston. During the working process, electrical and electromechanical time constants are usually used to determine the dynamic performance of EMLA. The specific expression is

$$\begin{cases} t_e = \frac{L}{R} \\ t_M = \frac{MR}{K_m^2} \end{cases} \tag{5}$$

where $K_m$ is the electromagnetic force coefficient; $M$ is the dynamic mass of the actuator; $R$ and $L$ are the resistance and inductance of coils; $t_e$ and $t_m$ are the electrical time constant and the electromechanical time constant, respectively. The specific actuator parameters are shown in Table 2.

**Table 2.** Parameters of brake-by-wire unit based on direct drive pump–valve cooperative.

| Components | Item | Value | Unit |
|---|---|---|---|
| Hydraulic pump | Plunger area | 28 | mm$^2$ |
| | Length of pump chamber | 16 | mm |
| EMLA | Coil resistance | 1.40 | Ω |
| | Coil inductance | 0.91 | mH |
| | Back EMF constant | 24.61 | Vs/m |
| | Electromagnetic force coefficient | 24.61 | N/A |
| Active valve | Valve core diameter | 4 | mm |
| | Valve hole diameter | 8 | mm |
| | Opening time | 2 | ms |

When the brake-by-wire unit based on direct drive pump–valve cooperative works, the two active valves connected with the pump chamber work alternately to cooperate with the adjusting pressure, keeping pressure, and releasing pressure mode in time. Therefore, the structural parameters of the active valve have an effect on the dynamic performance of the brake-by-wire unit based on the direct drive pump–valve cooperative.

Under the control signal, the valve needs to continuously achieve the opening and closing motion state in a short time, and the valve core stroke is short. Therefore, a ball valve type valve core with a simple structure, less wear, and a good sealing effect is adopted. Using an electromagnet as the core component of the active valve can shorten the opening and closing time of the valve core and increase its opening and closing frequency. A detailed analysis of the active valve can be found in reference [33]. The parameters of the selected active valve are shown in Table 2.

## 4. System Modeling

### 4.1. Modeling of the EMLA

The EMLA used in the brake-by-wire unit based on direct drive pump–valve cooperative is composed of the outer and inner yoke, the coil skeleton, the coil winding, and the permanent magnet array. The EMLA is directly connected to the hydraulic plunger, subject to friction, hydraulic resistance, etc. The coupling model of its mechanical, magnetic, and electrical subsystems is as follows:

$$\begin{cases} M\frac{d^2x}{dt^2} = F_m - F_f - P_1S_1 - F_{dis} \\ F_m = NB_el_eI = K_mI \\ u = IR + \frac{dI}{dt}L + K_ev \end{cases} \tag{6}$$

where $N$ is the coil turns; $B_e$ is the magnetic field strength; $l_e$ is the single turn coil length; $I$ is the coil current; $x$ is the displacement of the actuator; $F_f$ is the friction force affected by the actuator movement; $F_{dis}$ is the uncertainty error and interference; $u$ is the supply voltage; $K_e$ is the back EMF constant, and $v$ is the moving speed of the actuator. In order to improve the accuracy of modeling, a simple and effective expression of friction force is established, which is regarded as a static nonlinear function of velocity. The specific expression is as follows [34]:

$$F_f = B_1\dot{x} + A_f\arctan(\beta\dot{x}) \tag{7}$$

where $B_1$ is the viscosity coefficient; $A_f$ is the Coulomb friction coefficient, and the traditional symbolic function is represented by the smoothing function arctan. $\beta$ is a constant, and $\beta$ is set large enough so that the function retains the properties of a symbolic function and makes the expression of friction more realistic.

### 4.2. Hydraulic System Modeling

The direct drive pump needs to work under high-pressure conditions, so the compressibility of the liquid must be considered. To simplify the mathematical model of the direct drive pump, it is assumed that there is no pressure loss along the flow path or local pressure loss when the liquid is flowing; the direct drive pump chamber and hydraulic piston do not deform; the pressure in the pump chamber is equal, and the entire brake unit is well sealed with no hydraulic oil leakage [35]. The pressure change model of the direct drive pump is

$$\dot{P_1} = \beta_e \frac{S_1 \dot{x} - Q}{S_1(l - x)} \tag{8}$$

where $\beta_e$ is the effective bulk elastic modulus of hydraulic oil; $Q$ is the flow rate of direct drive pump, and $l$ is the direct drive pump chamber length.

The active valve plays the role of flow distribution in the brake-by-wire unit based on direct drive pump–valve cooperative. Assuming that the active valve is fully open, the pressure difference between the two ends of the active valve spool is small, and the flow rate of the valve hole is simplified as follows:

$$Q = k_0 c_0 A_0 \sqrt{\frac{2}{\rho} \Delta P} \tag{9}$$

where $k_0$ is the flow linearization coefficient; $c_0$ is the flow coefficient of the valve port; $A_0$ is the valve port area; $\Delta P$ is the hydraulic pressure difference in the valve port, and $\rho$ is the hydraulic oil density.

The brake wheel cylinder realizes the conversion of hydraulic pressure to braking force. Due to the small displacement of the piston $x_h$ and the fact that the piston displacement remains basically unchanged when the pressure of the wheel cylinder increases, $\dot{x}_h$ can be ignored. The pressure change model of the wheel cylinder is

$$\dot{P_2} = \beta_e \frac{Q}{S_2(l_h + x_h)} \tag{10}$$

where $l_h$ is the length of the wheel cylinder chamber.

### 4.3. Quarter-Car Dynamics Model Establishment

This paper needs to study the slip rate control under emergency braking so a 1/4 Vehicle dynamics model is established. The vehicle longitudinal dynamics model is

$$F_x = -m_c \dot{v}_c = \mu_c m_c g \tag{11}$$

where $F_x$ is the ground braking force; $m_c$ is a quarter of the vehicle's weight; $v_c$ is the tire's longitudinal speed, and $\mu_c$ is the longitudinal friction coefficient of the tire. The wheel dynamics model is

$$J\dot{\omega} = F_x r_b - T_b \tag{12}$$

where $T_b$ is the braking torque, and $J$ is the wheel's moment of inertia. For the tire longitudinal friction coefficient $\mu$, the Burckhardt tire model is used to calculate

$$\mu(s) = c_1[1 - \exp(-c_2 s)] - c_3 s \tag{13}$$

where $s$ is slip rate, and $c_1$, $c_2$, and $c_3$, respectively, represent the peak parameter, shape parameter, and difference parameter of the road adhesion coefficient curve corresponding to the typical road surface.

### 4.4. Co-Simulation Model Construction

The brake-by-wire unit based on direct drive pump–valve cooperative is a nonlinear time-varying system with mechanical, electrical, magnetic, and hydraulic coupling. Considering the necessary requirements of system testing and actual working conditions, AMESim is used to model the direct drive pump, active valve, wheel cylinder, hydraulic pipes, and other components. At the same time, the EMLA and the controller model are established in Matlab/Simulink. The co-simulation model is shown in Figure 2. Based on the interface technology between AMESim and Simulink, a co-simulation interface is established in AMESim, and corresponding S-functions are set in Simulink, fully utilizing the advantages of AMESim in nonlinear dynamic hydraulic system modeling and powerful functions of Matlab in complex controller mathematical model building and data calculation processing, to obtain fast and real-time results.

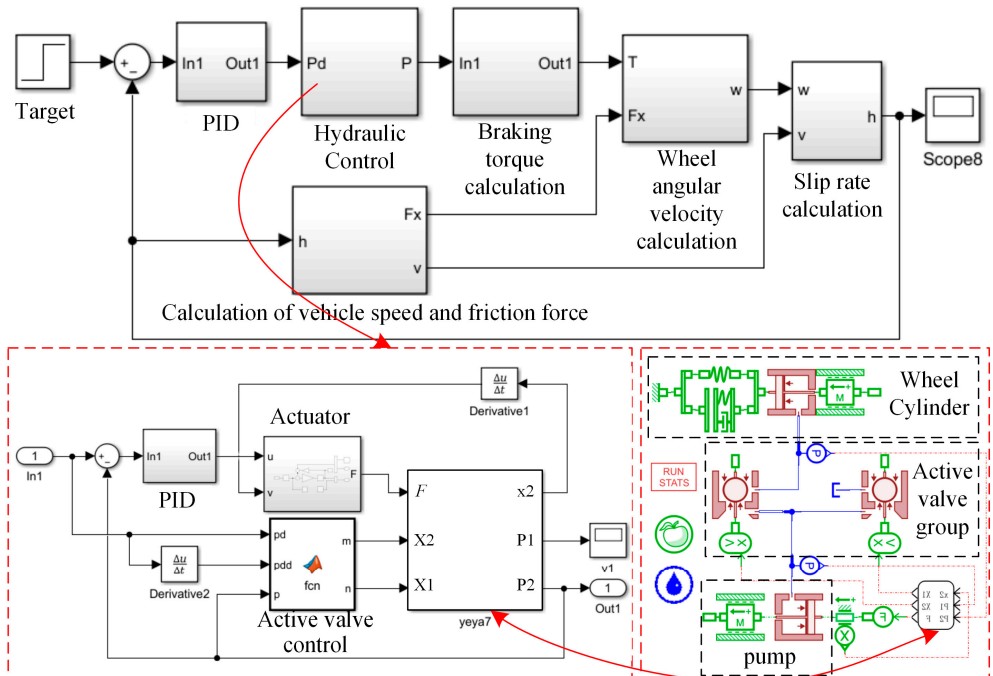

**Figure 2.** Electro-hydraulic co-simulation model.

## 5. Results and Analysis

### 5.1. The Experimental Platform

The experimental platform of the brake-by-wire unit based on direct drive pump–valve cooperative, as shown in Figure 3, is established. The personal computer transmits the control signal to the controller. Then, the controller controls the power drive module to control the driving voltage of EMLA and the switch of the active valve, which controls the pressure of the brake cylinder. At the same time, the pressure sensor and the position sensor collect the signal and transmit it to the personal computer through the controller, which forms the signal feedback. The controller adopts RTU-BOX, a rapid control prototype system, and its digital controller adopts TMS320C28346, a 32-bit floating point digital signal processor with a main frequency of 300 MHz. The position sensor with 0.01 mm resolution provides the position feedback of the electromagnetic linear actuator, and the pressure sensor with 0.01 MPa resolution provides the hydraulic pressure feedback of the wheel cylinder.

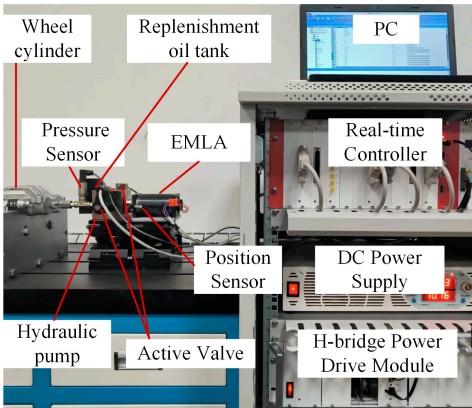

**Figure 3.** The experimental platform.

### 5.2. Analysis of Hydraulic Pressure Response Performance

We simulated the emergency braking condition of the target vehicle and set the step response target as 4 MPa. The test results are shown in Figure 4. Under the PID control algorithm, the brake unit can quickly achieve stability without overshooting, and the response time of simulation and experiment is 0.023 s and 0.026 s, respectively. The simulated curve and response time are close to experimental results, which verifies the validity of the co-simulation model.

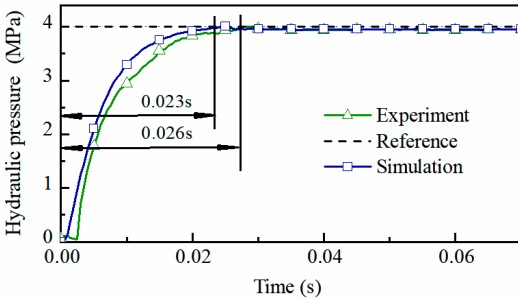

**Figure 4.** Step response curve.

The experimental results of the EMLA and active valves under the PID controller are shown in Figure 5. When the system reaches the target pressure, and the working mode is switched to the pressure-maintaining mode, the pressure-maintaining valve closes, causing small fluctuations in the hydraulic pressure. At the same time, the actuator control signal is 0, and the current value rapidly decreases. Under the pressure in the pump chamber, the actuator moves toward the reset direction for a period of displacement.

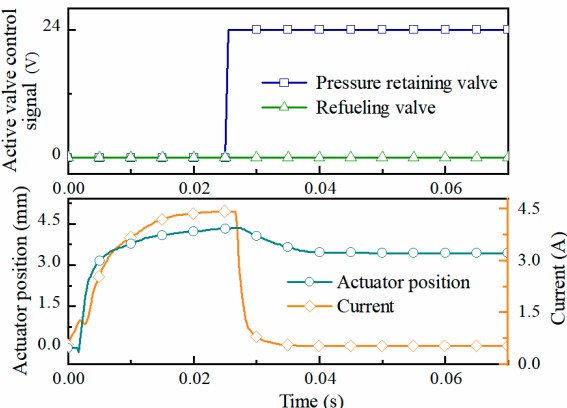

**Figure 5.** Status of the actuator and the active valves.

*5.3. Analysis of Hydraulic Pressure Tracking Performance*

To verify the hydraulic pressure tracking performance of the brake-by-wire unit based on direct drive pump–valve cooperative, the target sinusoidal pressure signal frequency is set to 2.5 Hz and amplitude to 2.5 MPa, and the hydraulic pressure tracking curve and the system state in the experiment are shown in Figure 6. PID control can track a sinusoidal signal well. In the process of sine wave target tracking, the average error of simulation and experiment PID control is 0.197 MPa and 0.218 MPa, and the amplitude attenuation of simulation and experiment PID control is 0.106 MPa and 0.121 MPa, respectively. The results show that the braking unit can effectively control the hydraulic pressure under a sine wave target. However, due to the limitation of DC power supply voltage and sensor measurement accuracy, as well as the difficulty of system oil filling in actual conditions, the actual pressure curve differs from the simulation.

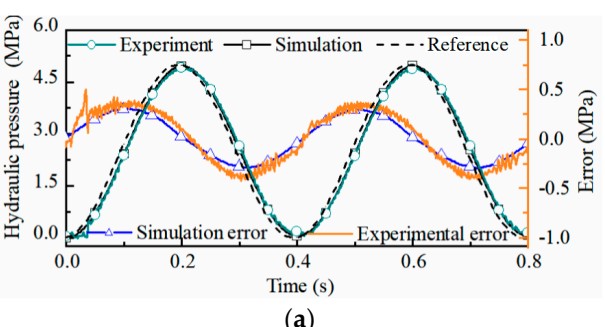
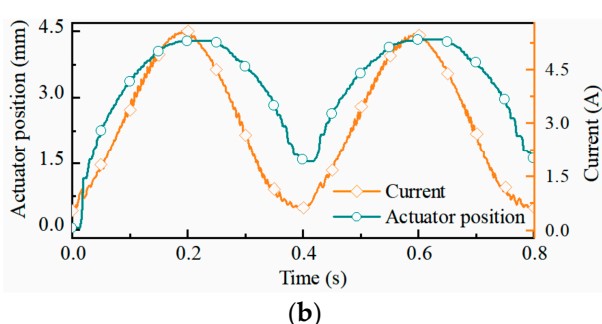

(**a**)  (**b**)

**Figure 6.** Experimental and simulation results under the sinusoidal target: (**a**) Hydraulic pressure tracking curve for simulation and experiment; (**b**) The system state in the experiment.

To further simulate the pressure tracking performance, we set the target signal of derivative mutation; the target hydraulic pressure frequency of the triangular wave braking was set to 2.5 Hz, and the amplitude to 2.5 MPa. The hydraulic pressure tracking curve and the system state in the experiment are shown in Figure 7. At 0.2 s and 0.4 s, the derivative of the target signal suddenly changes, but the PID controller can still track the triangular wave signal well. Due to the hydraulic oil in the system not being fully filled, the actuator position at 0.4 s is higher than the initial state. In the process of triangular wave target tracking, the average error of simulation and experiment PID control is 0.141 and 0.203, respectively. The results show that the proposed brake unit has good accuracy when maintaining a certain rate of increasing or decreasing hydraulic pressure, which verifies its good tracking performance.

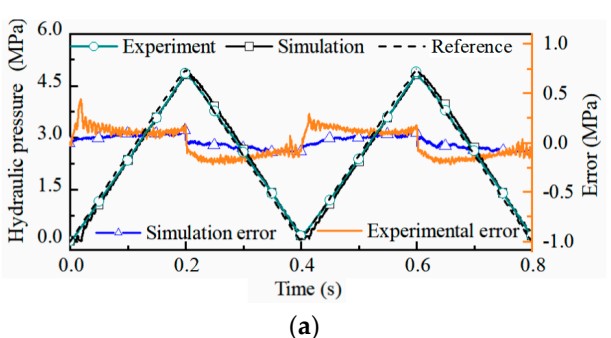
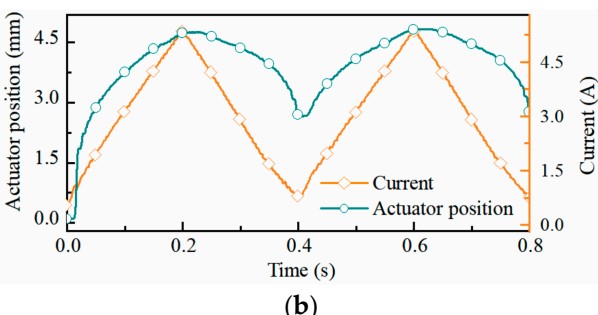

(**a**)  (**b**)

**Figure 7.** Experimental and simulation results under the triangular wave target: (**a**) Hydraulic pressure tracking curve for simulation and experiment; (**b**) The system state in the experiment.

*5.4. Analysis of the Impact of Parameters on System Performance*

The structure parameters that can be optimized were analyzed to obtain the influence trend of parameter changes on the system performance. The average tracking error

and amplitude attenuation of the sinusoidal signal were used to express the influence of structural parameters on system performance changes. For this purpose, the change degree curves of independent variables are drawn in this paper, as shown in Figures 8–10. Through the curve changes, the impact of parameter changes on system performance is intuitively shown.

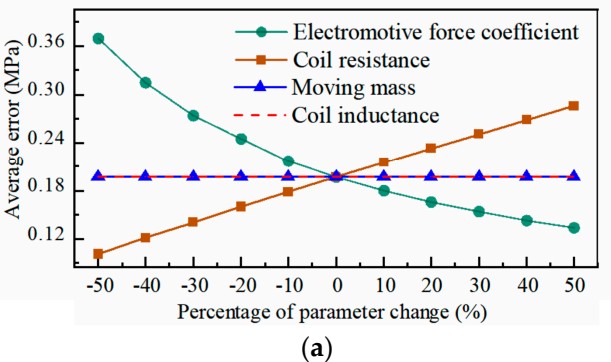 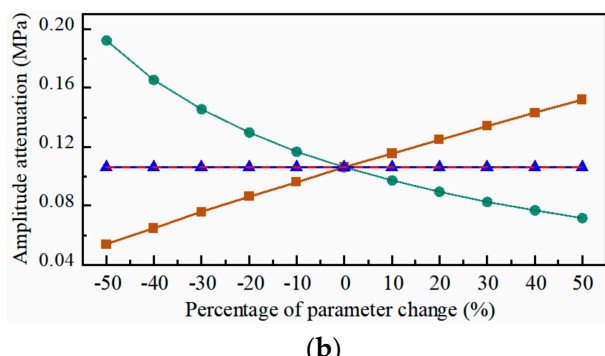

(a)　　　　　　　　　　　　　　　　　　(b)

**Figure 8.** The influence of actuator parameters on system performance: (**a**) The influence on average error; (**b**) The influence on amplitude attenuation.

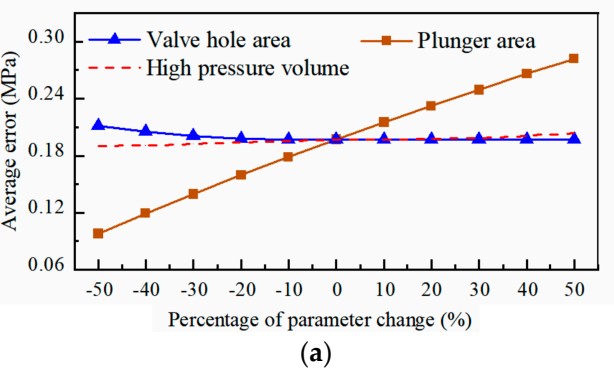 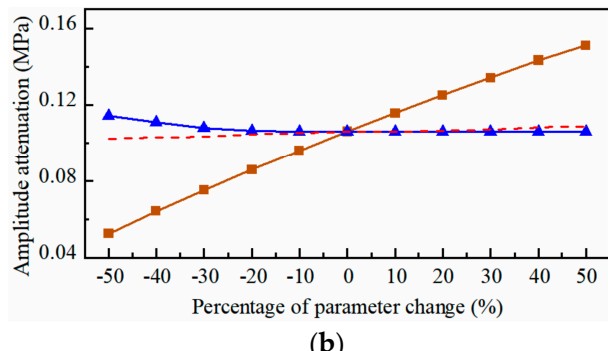

(a)　　　　　　　　　　　　　　　　　　(b)

**Figure 9.** The influence of hydraulic system parameters on system performance: (**a**) The influence on average error; (**b**) The influence on amplitude attenuation.

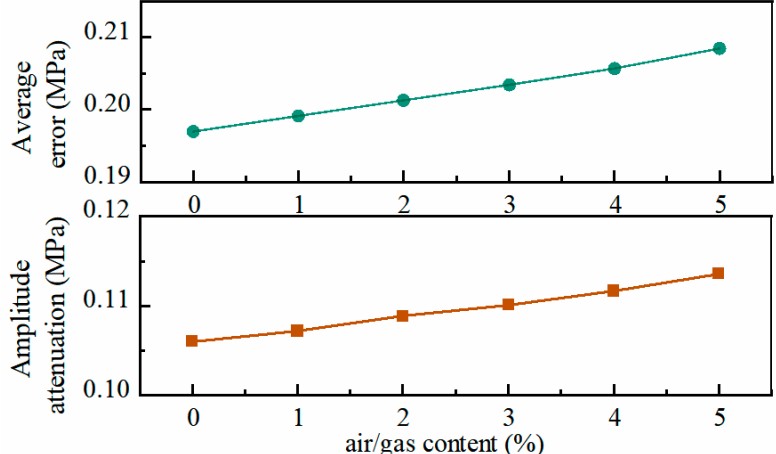

**Figure 10.** The influence of air content on average error and amplitude attenuation.

5.4.1. Actuator Parameter

As can be seen from Figure 8, for moving mass, including the mass of the actuator and the mass of the plunger inside the pump, although reducing the moving mass can reduce the load on the actuator, compared to the load on the actuator caused by hydraulic

pressure, the smaller dynamic mass has a smaller impact on the system response speed and tracking accuracy. Therefore, there is no need to lightweight the dynamic mass of the actuator. For the electromagnetic force coefficient, an increase in the electromagnetic force coefficient will increase the electromagnetic force under the same current, and the target tracking and response can be carried out with a smaller current for the same hydraulic load, which will improve the precision of pressure control. Therefore, the electromagnetic force coefficient should be increased under the premise of a comprehensive consideration of installation space and cost. For the resistance, as the controller signal is used as the input voltage of the system, the smaller the resistance under the same voltage, the larger the current, and the larger the electromagnetic force under the same actuator state. Under the same hydraulic load, a smaller control signal is needed to track the target, which will improve the pressure control accuracy. For inductance, because the inductance term in Equation (6) has a smaller value than the other two terms, the change in inductance makes the change in current smaller under the same voltage, so the pressure control accuracy is almost unchanged.

### 5.4.2. Hydraulic System Parameter

As can be seen from Figure 9, for the plunger area, when the plunger area is reduced, the same electromagnetic force can generate greater hydraulic pressure. However, when the area is too small, the actuator displacement increases under the same hydraulic pressure target. So, it is necessary to reduce the plunger area while considering the actuator stroke. For the high-pressure volume, the increase in the high-pressure volume will result in a slower system pressure response and a decrease in pressure control accuracy. Additionally, it will also increase the actuator movement displacement, which limits the actuator stroke. Therefore, the hydraulic pipeline should be reduced as much as possible to reduce the high-pressure volume. For the valve port area, an increase in the valve port area will increase the instantaneous flow rate, but if it is too large, it will cause severe flow oscillation, resulting in excessive hydraulic pressure error in the keeping pressure mode, so the valve port area can be increased without increasing the error.

As can be seen from Figure 10, the elastic modulus of the oil is a comprehensive performance parameter that is related to factors such as gas content in the oil, system pressure, and oil temperature. When the oil is mixed with 1% air, the bulk elastic modulus will drop to about 5% of the pure oil. The pressure control precision of the system can be increased by increasing the bulk elastic modulus of the oil. Therefore, hydraulic oil with a large bulk elastic modulus should be selected, and measures should be taken to prevent air from being mixed into the oil.

### 5.5. ABS Performance Analysis

It is known from the above section that the proposed brake-by-wire unit based on direct drive pump–valve cooperative in this paper responds quickly and can accurately control the pressure in the cylinder, making it easier to realize the anti-lock braking function. This section further conducts a simulation study on the anti-lock braking performance of the brake-by-wire unit based on direct drive pump–valve cooperative and sets the initial speed at 100 km/h. Dry asphalt ($c_1$ = 1.280, $c_2$ = 23.99, $c_3$ = 0.52) is selected for the road surface. According to the formula, when the road surface adhesion coefficient $\mu$ derivative is zero, the function is at the maximum point, and the slip rate here is the optimal slip rate

$$s_{opt} = \frac{1}{c_2} \ln \frac{c_1 c_2}{c_3} \tag{14}$$

The optimal slip rate is 0.17, and the wheel anti-lock braking system starts when the simulation starts. The simulation results are shown in Figure 11. As the brake unit can accurately control the wheel cylinder pressure, it can accurately control the wheel slip rate. The response time of the slip rate is 0.422 s; the speed needs 2.404 s from the initial

speed of 100 km/h to the stop, and the braking distance is 34 m, which meets the braking safety requirements.

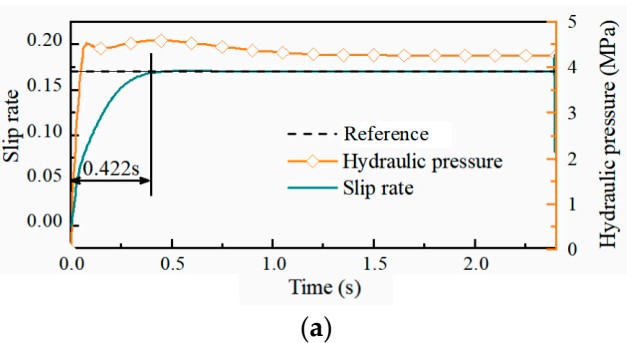 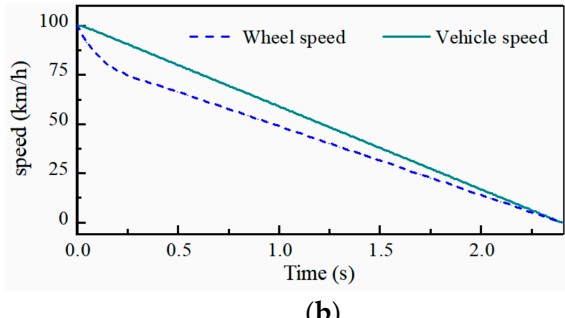

(**a**)                                                   (**b**)

**Figure 11.** Simulation results of slip rate control: (**a**) Slip rate and hydraulic pressure curve; (**b**) Vehicle speed and wheel speed curve.

## 6. Conclusions

This paper proposes a brake-by-wire unit based on a direct drive pump–valve cooperative, which has the advantages of rapid response and no deterioration of wheel side space and unsprung mass. The co-simulation model of the brake unit is established based on the Simulink-AMESim co-simulation. The pressure experiment shows that it has a good pressure control performance, and the experimental curve is close to the simulation curve, which proves the accuracy of the simulation model. The feasibility of the simulation is further verified by ABS simulation, which provides a new implementation scheme for automotive distributed braking system. The characteristics of the key parameters are analyzed, and the influence of each parameter on the control performance is researched by changing the percentage of parameter change. This lays the foundation for further research. The structure optimization of EMLA and the design of the hydraulic control algorithm are the keys to improving the performance of brake units and also the main research direction in the future.

**Author Contributions:** Conceptualization: P.Y. and C.T.; methodology: P.Y., Z.S. and C.T.; software: P.Y. and H.X.; formal analysis: P.Y. and C.T.; supervision: C.T.; writing—original draft preparation: P.Y.; writing—review and editing: P.Y., Z.S., Y.R., H.X. and C.T. All authors have read and agreed to the published version of the manuscript.

**Funding:** This work was supported by Shandong Natural Science Foundation Project (Grant No. ZR2023ME178), Open Fund of State Key Laboratory of Automobile Simulation and Control of Jilin University (Grant No. 20210224), Shandong Province Science and Technology Small and Medium Enterprises Innovation Ability Enhancement Project (Grant No. 2023TSGC0404), Innovation team project of "Qing-Chuang science and technology plan" of colleges and universities in Shandong Province (Grant No. 2019KJB027 and Grant No.2022KJ232) and Young Technology Talent Supporting Project of Shandong Province (Grant No. SDAST2021QT20).

**Data Availability Statement:** Not applicable.

**Conflicts of Interest:** The authors declare no conflict of interest.

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
