# Peer review of "Design and Analysis of Brake-by-Wire Unit Based on Direct Drive Pump–Valve Cooperative"

_actuators, doi:10.3390/act12090360_

Round 1

Reviewer 1 Report

The study is very interesting and will have application in distributed braking systems. However, the manuscript needs to undergo the following revisions:

1. References are not enough, and the author must cite at least 35 to 40 references. There is a lot of literature on distributed braking systems. Please cite at least 35 recently published articles.

2. The paper is poorly written from a language point of view. The sentence structure is not good, and it is hard to understand the meaning of the paragraph.

3. There are some simulation parameters in Table 1, Table 2, and Table 3. Are these parameters from published literature or the manufacturer? Please cite the references for them.

4. Please combine Tables 1 to 3 into a single Table for simulation parameters.

5. Novelty is not clearly mentioned in the paper

6. Abstract is poorly written. Rewrite the abstract by focusing on research problems, the basic design of the study, your design methodology, and major findings or trends.

7. The introduction section should be structured in a way that the reader should be able to:

a. Understand the overall topic;

b. Understand the existing gap;

c. Understand how this paper is going to contribute to this gap;

8. Conclusions are poorly written. Rewrite conclusions. Wrap up your ideas and leave the reader with a strong final impression. Please restate the problem statement addressed in the paper, summarise your overall arguments or findings, and suggest the key takeaways from your paper.

9. Figure 3 needs more explanation

10. Figure 6, Figure 7, and Figure 8 shows that the quality of the image appearance is not good. images are blurry.

11. The mathematical model is built from equation 7. Please cite the reference for the equation.

The paper is difficult to understand from the language point of view.

Author Response

Reply to the comments

Dear Editor,

Thank you for your letter and the valuable feedback provided by the reviewers. We sincerely appreciate their guidance and input.

We have prepared a revised version of the manuscript titled "Design and analysis of Brake-by-wire Unit Based on Direct Drive Pump-Valve Cooperative" for your reconsideration. All co-authors have carefully reviewed and approved the revised manuscript.

In response to the reviewers' comments, we have made revisions to address the concerns raised. Each point raised by the reviewers has been thoroughly addressed in the revised manuscript, and we have also made necessary corrections, and content revisions are marked in red and language revisions are marked in blue for easy identification.

We would like to express our gratitude to the reviewers for their recognition and guidance on our research. Their feedback has been invaluable in improving the quality and clarity of our work. We hope that the revised manuscript now meets the requirements of the journal.

Best regards,

Cao Tan

Reply to the comments of Reviewer: 1

Comments to the Author

The study is very interesting and will have application in distributed braking systems. However, the manuscript needs to undergo the following revisions.

Reply:

Thank you very much for your professional review of our article. We sincerely appreciate the concerns and questions you raised.

We take your suggestions very seriously and we have taken them seriously and extensively revised our previous manuscripts, and these changes have been important in improving the quality and accuracy of the article. Below are responses to each point of your comments.

1.Comment: References are not enough, and the author must cite at least 35 to 40 references. There is a lot of literature on distributed braking systems. Please cite at least 35 recently published articles.

  1. Reply:

Thanks for your valuable suggestion. We have added the following references.

[16] Picasso, B.; Caporale, D.; Colaneri, P. Braking control in railway vehicles: A distributed preview approach. IEEE Transactions on Automatic Control 2017, 63, 189-195.

[22] Yu, D.; Wang, W.; Zhang, H.; Xu, D. Research on anti-lock braking control strategy of distributed-driven electric vehicle. IEEE Access 2020, 8, 162467-162478.

[23] Henderson, J. P.; Plummer, A.; Johnston, N. An electro-hydrostatic actuator for hybrid active-passive vibration isolation. International Journal of Hydromechatronics 2018, 1, 47-71.

[30] Ji, Y.; Zhang, J.; Zhang, J.; He, C.; Hou, X.; Han, J. Constraint performance slip ratio control for vehicles with distributed electrohydraulic brake-by-wire system. Proceedings of the Institution of Mechanical Engineers, Part D: Journal of Automobile Engineering 2023, 09544070231157154.

[32] Wu, Y.; Liu, M.; Zhang, G.; Hou, Z.; Lin, Q.; Yuan, H. Study on high frequency response characteristics of a moving-coil-type linear actuator using the coils combinations. International Journal of Hydromechatronics, 2022, 5, 226-242.

[34] Chen, Z.; Yao, B.; Wang, Q. Accurate motion control of linear motors with adaptive robust compensation of nonlinear electromagnetic field effect. IEEE/ASME Transactions On Mechatronics 2012 18, 1122-1129.

2.Comment: The paper is poorly written from a language point of view. The sentence structure is not good, and it is hard to understand the meaning of the paragraph.

  1. Reply:

Thanks for your valuable suggestion. We have revised the sentence structure of the paper, and the specific changes have been highlighted in blue font.

3.Comment: There are some simulation parameters in Table 1, Table 2, and Table 3. Are these parameters from published literature or the manufacturer? Please cite the references for them.

  1. Reply:

Thanks for your valuable suggestion. These parameters are designed according to the braking requirements of the drive-by-wire chassis. The specific design ideas are as follows

According to the maximum braking required by the wheel, the corresponding wheel cylinder pressure is determined. Firstly, the appropriate Plunger area in the pump is selected by comprehensively considering the magnification and axial size. Secondly, the actuator parameters are determined through the electromagnetic force required and electrical and electromechanical time constants. Finally, the active valve parameters are selected by considering the influence of the active valve structure parameters on the dynamic performance of the brake unit.

4.Comment: Please combine Tables 1 to 3 into a single Table for simulation parameters.

  1. Reply:

Thanks for your valuable suggestion. We have combined the structural parameters of the brake unit into the parameter table shown below.

Table 2. Parameters of brake-by-wire unit based on direct drive pump-valve cooperative.

components

item

value

unit

Hydraulic pump

Plunger area

28

mm2

Length of pump chamber

16

mm

Electromagnetic linear actuator

Coil resistance

1.40

Ω

Coil inductance

0.91

mH

Back EMF constant

24.61

Vs/m

Electromagnetic force coefficient

24.61

N/A

Active valve

Valve core diameter

4

mm

Valve hole diameter

8

mm

Opening time

2

ms

5.Comment: Novelty is not clearly mentioned in the paper

  1. Reply:

Thanks for your valuable suggestion. The main contributions of the paper are revised as follows

The main contributions are summarized as follows. (1) A brake-by-wire unit based on direct drive pump-valve cooperative is proposed, which has the advantages of rapid response and no deterioration of wheel side space and unsprung mass. (2) Based on the Simulink-AMESim co-simulation platform, a co-simulation model of the brake unit is established and verified. (3) The characteristics of the key parameters are analyzed, and the influence law of each parameter on the control performance is investigated. The rest of this work is organized as follows. Parameter matching design of the key parameters is carried out in Section 2. The dynamic model of the brake-by-wire unit based on direct drive pump-valve cooperative is established in Section 3. The feasibility is verified by pressure experiments and ABS simulations in Section 4. Finally, conclusion is made in Section 5.

6.Comment: Abstract is poorly written. Rewrite the abstract by focusing on research problems, the basic design of the study, your design methodology, and major findings or trends.

  1. Reply:

Thanks for your valuable suggestion. We have revised the abstract of the paper as follows

Aiming at the requirements of distributed braking and advanced automatic driving, a brake-by-wire unit based on direct drive pump-valve cooperative is designed, to realize the wheel cylinder pressure regulation, the hydraulic pump directly driven by the electromagnetic linear actuator coordinates with the active valve. which has the advantages of rapid response and no deterioration of wheel side space and unsprung mass. Firstly, by analyzing the working characteristics and braking performance requirements of the braking unit, the key parameters of the system are matched. Then, In order to ensure the accuracy of the simulation model, the co-simulation model of the brake unit is established based on the Simulink-AMESim co-simulation platform, and the influence law of key parameters on the control performance is analyzed. Finally, the experimental platform of the brake unit is built. The accuracy of the co-simulation model and the feasibility of the brake-by-wire unit based on direct drive pump-valve cooperative are verified through the pressure control experiment and ABS simulation. And it shows that the braking unit has good dynamic response and steady-state tracking effect.

7.Comment: The introduction section should be structured in a way that the reader should be able to:

  1. Understand the overall topic;
  2. Understand the existing gap;
  3. Understand how this paper is going to contribute to this gap;
  4. Reply:

Thanks for your valuable suggestion. We have revised the introduction of the paper as follows

With the development of vehicle chassis towards the direction of X-by-wire chassis and slide chassis, the demand for distributed braking system is becoming more and more intense. At the same time, high-level automatic driving puts forward higher requirements for the response speed and control accuracy of the drive-by-wire system [1-4].

Electronic Hydraulic Brake (EHB) System has been widely favored due to its advantages such as rapid response, high power density and brake structure compatibility [5-9]. In 2013, Bosch launched I-Booster, a motor servo booster that is independent of vacuum booster, which is a typical motor servo EHB system. TRW introduced a high-pressure accumulator-based EHB system called the SCB, which has a brake master cylinder consisting of front and rear chambers and parallel pistons. Since the pistons can be moved fore and aft and the front part is connected to the front wheel, the pressure of the system can be adjusted through the rear part of the pistons. [10]. In 2010, Hitachi launched the EHB system named e-ACT, which uses an electric motor to drive an actuator that pushes a master cylinder piston, and the rotational force of the motor is converted into linear motion by a ball screw. [11]. In 2021, Continental launched a new generation of EHB system MKC2 based on MKC1, which uses a multi-logic framework with independent partitions, successfully reducing the number of brake system components, and its modular and scalable design provides ideas for future dynamic control of brake systems.

At present, EHB system is still the mainstream of the development of brake-by-wire system. In order to take into account the needs of market and efficiency, it is of great significance to explore a new brake-by-wire structure to improve the performance of brake-by-wire and simplify the system structure [12-16]. Sun designed and developed an integrated master cylinder and decoupling electro-hydraulic composite braking system, and proposed the hydraulic braking force distribution strategy [17]. Wu designed an integrated EHB system. By controlling the solenoid valve and designing the working mode switching control strategy, the working mode switching was realized to achieve high redundancy and independent control of four wheel cylinders of the braking system [18]. EHB with different characteristics have been proposed to promote the development of brake-by-wire technology, but it is difficult to directly apply to distributed braking systems [19-23]. It is necessary to design a distributed brake-by-wire unit to meet the needs of intelligent driving, and meet the requirements of new energy vehicles for precise control of braking force and lightweight. Fu designed a new type of electromechanical brake with automatic wear adjustment function, and adopted worm gear mechanism for power transmission, which has the advantages of low delay and small size [24]. Yu designed a distributed electro-hydraulic brake-by-wire system, which uses a motion conversion mechanism to change the motion of the rotating motor into linear motion and then drive the hydraulic wheel cylinder [25]. Chang designed an EHB unit that uses a linear motor to drive the hydraulic piston, amplifying the driving force of the motor through unequal diameter hydraulic pistons and wheel cylinder pistons [26]. The existing EHB are often improved based on proven hydraulic braking systems, which still retain complex hydraulic pipelines [27-30]. Distributed brake-by-wire systems often require the entire brake unit to be integrated into the caliper, deteriorating wheel side space and unsprung mass. Therefore, This paper designed a brake-by-wire unit based on direct drive pump-valve cooperative, through the electromagnetic linear actuator directly driven by the hydraulic pump and the active valve coordination to achieve wheel cylinder pressure regulation.

The main contributions are summarized as follows. (1) A brake-by-wire unit based on direct drive pump-valve cooperative is proposed, which has the advantages of rapid response and no deterioration of wheel side space and unsprung mass. (2) Based on the Simulink-AMESim co-simulation platform, a co-simulation model of the brake unit is established and verified. (3) The characteristics of the key parameters are analyzed, and the influence law of each parameter on the control performance is investigated. The rest of this work is organized as follows. Parameter matching design of the key parameters is carried out in Section 2. The dynamic model of the brake-by-wire unit based on direct drive pump-valve cooperative is established in Section 3. The feasibility is verified by pressure experiments and ABS simulations in Section 4. Finally, conclusion is made in Section 5.

8.Comment: Conclusions are poorly written. Rewrite conclusions. Wrap up your ideas and leave the reader with a strong final impression. Please restate the problem statement addressed in the paper, summarise your overall arguments or findings, and suggest the key takeaways from your paper.

  1. Reply:

Thanks for your valuable suggestion. We have revised the conclusion as follows

This paper proposes a brake-by-wire unit based on direct drive pump-valve cooperative, which has the advantages of rapid response and no deterioration of wheel side space and unsprung mass. The co-simulation model of the brake unit is established based on the Simulink-AMESim co-simulation. The pressure experiment shows that it has good pressure control performance, and the experimental curve is close to the simulation curve, which proves the accuracy of the simulation model. The feasibility of the simulation is further verified by ABS simulation, which provides a new implementation scheme for automotive distributed braking system. The characteristics of the key parameters are analyzed, and the influence of each parameter on the control performance is researched by changing the percentage of parameter change. This lays the foundation for further research. The structure optimization of electromagnetic linear actuator and the design of hydraulic control algorithm are the key to improve the performance of brake unit, and also the main research direction in the future.

9.Comment: Figure 3 needs more explanation

  1. Reply:

Thanks for your valuable suggestion. The interpretation of Figure 3 is revised as follows

The personal computer transmits the control signal to the controller, and then controls the power drive module to control the driving voltage of EMLA and the switch of the active valve, which controls the pressure of the brake cylinder. At the same time, the pressure sensor and the position sensor collect the signal and transmit it to the personal computer through the controller, which forms the signal feedback. The controller adopts RTU-BOX, a rapid control prototype system, and its digital controller adopts TMS320C28346, a 32-bit floating point digital signal processor with a main frequency of 300MHz. The position sensor with 0.01mm resolution provides the position feedback of the electromagnetic linear actuator, and the pressure sensor with 0.01MPa resolution provides the hydraulic pressure feedback of the wheel cylinder.

10.Comment: Figure 6, Figure 7, and Figure 8 shows that the quality of the image appearance is not good. images are blurry.

  1. Reply:

Thank you for your valuable suggestion. We are sorry for the mistake, and the modification figure is as follows

(a)

(b)

Figure 6. Experimental and simulation results under the sinusoidal target: (a) Hydraulic pressure tracking curve for simulation and experiment; (b) the system state in the experiment.

(a)

(b)

Figure 7. Experimental and simulation results under the triangular wave target: (a) Hydraulic pressure tracking curve for simulation and experiment; (b) the system state in the experiment.

(a)

(b)

Figure 8. The influence of actuator parameters on system performance: (a) The influence on average error; (b) The influence on amplitude attenuation.

11.Comment: The mathematical model is built from equation 7. Please cite the reference for the equation.

  1. Reply:

Thanks for your valuable suggestion. Add the reference below

Chen, Z.; Yao, B.; Wang, Q. Accurate motion control of linear motors with adaptive robust compensation of nonlinear electromagnetic field effect. IEEE/ASME Transactions On Mechatronics 2012 18, 1122-1129.

Reviewer 2 Report

This study focuses on modeling and analyzing an electro-hydraulic brake unit. According to the reviewer, the manuscript is well-structured and engaging in its subject matter. The reviewer's comments are provided below:

1.It is suggested to provide a concise summary of the contributions of this study within the introduction, while also emphasizing the significance of the analysis conducted on the electro-hydraulic brake (EHB) unit.

2.To enhance readability, the authors might consider avoiding lengthy sentence constructions. For instance, the initial sentence of the abstract could be revised.

3.In addition to displaying the schematic diagram of the brake-by-wire unit as depicted in Figure 1, it would be beneficial to include a standard schematic of the pump-valve cooperative hydraulic system. This addition will offer a clearer depiction of the hydraulic system.

4. In the section of the experimental platform, it is recommended to provide information about the specific brands and models of valves and pumps that were chosen.

5.In the conclusion section, it is advised to expand the discussion to encompass both the simulation and experimental results, providing a comprehensive overview of the findings.

To enhance readability, the authors might consider avoiding lengthy sentence constructions. For instance, the initial sentence of the abstract could be revised.

Author Response

Reply to the comments of Reviewer: 2

Comments to the Author

This study focuses on modeling and analyzing an electro-hydraulic brake unit. According to the reviewer, the manuscript is well-structured and engaging in its subject matter. The reviewer's comments are provided below.

Reply:

Thank you very much for your professional review of our article. We sincerely appreciate the concerns and questions you raised.

We take your suggestions very seriously and we have taken them seriously and extensively revised our previous manuscripts, and these changes have been important in improving the quality and accuracy of the article. Below are responses to each point of your comments.

1.Comment: It is suggested to provide a concise summary of the contributions of this study within the introduction, while also emphasizing the significance of the analysis conducted on the electro-hydraulic brake (EHB) unit.

  1. Reply:

Thanks for your valuable suggestion. We have summarized the major contributions at the end of the introduction.

The main contributions are summarized as follows. (1) A brake-by-wire unit based on direct drive pump-valve cooperative is proposed, which has the advantages of rapid response and no deterioration of wheel side space and unsprung mass. (2) Based on the Simulink-AMESim co-simulation platform, a co-simulation model of the brake unit is established and verified. (3) The characteristics of the key parameters are analyzed, and the influence law of each parameter on the control performance is investigated. The rest of this work is organized as follows. Parameter matching design of the key parameters is carried out in Section 2. The dynamic model of the brake-by-wire unit based on direct drive pump-valve cooperative is established in Section 3. The feasibility is verified by pressure experiments and ABS simulations in Section 4. Finally, conclusion is made in Section 5.

2.Comment: To enhance readability, the authors might consider avoiding lengthy sentence constructions. For instance, the initial sentence of the abstract could be revised.

  1. Reply:

Thanks for your valuable suggestion. We have revised the sentence structure of the paper, and the specific changes have been highlighted in blue font.

3.Comment: In addition to displaying the schematic diagram of the brake-by-wire unit as depicted in Figure 1, it would be beneficial to include a standard schematic of the pump-valve cooperative hydraulic system. This addition will offer a clearer depiction of the hydraulic system.

  1. Reply:

Thank you for your valuable suggestion. We have modified the description and Figure 1 according to your suggestion. Specific modifications are as follows

The typical working process of the brake-by-wire unit based on direct drive pump-valve cooperative is divided into adjusting pressure mode, keeping pressure mode and release pressure mode, as shown in Figure 1. In the release pressure mode, the pressure maintaining valve is opened and the oil refill valve is closed. Under the action of brake wheel cylinder pressure and brake wheel cylinder return spring, the brake wheel cylinder hydraulic flow back hydraulic pump, actuator and hydraulic pump piston return to the initial position. If linear actuator does not return to the initial position due to hydraulic oil leakage or other reasons, the pressure maintaining valve is closed, the oil refill valve is opened, and the actuator drives the hydraulic pump piston back to the initial position. In the adjusting pressure mode, the pressure maintaining valve is opened and the oil refill valve is closed. The electromagnetic linear actuator drives the hydraulic piston, quickly discharging hydraulic oil from the hydraulic pump head into the brake wheel cylinder, pushing the brake wheel cylinder piston to eliminate brake clearance, and then quickly adjusting the brake wheel cylinder pressure by controlling the output force of the linear actuator. In the keeping pressure mode, the pressure maintaining valve and oil refill valve are closed, the actuator does not work and brake wheel cylinder pressure is maintained. The brake unit relieves the working burden of the electromagnetic linear actuator and reduces the working energy consumption under the keeping pressure mode.

(a)

(b)

Figure 1. The structure of brake-by-wire unit based on direct drive pump-valve cooperative: (a) The schematic diagram; (b) the prototype.

4.Comment: In the section of the experimental platform, it is recommended to provide information about the specific brands and models of valves and pumps that were chosen.

  1. Reply:

Thanks for your valuable suggestion. The valves and pump are designed according to the braking performance requirements of X-by-wire chassis, and the relevant technical parameters are added in the paper.

components

item

value

unit

Hydraulic pump

Plunger area

28

mm2

Length of pump chamber

16

mm

Electromagnetic linear actuator

Coil resistance

1.40

Ω

Coil inductance

0.91

mH

Back EMF constant

24.61

Vs/m

Electromagnetic force coefficient

24.61

N/A

Active valve

Valve core diameter

4

mm

Valve hole diameter

8

mm

Opening time

2

ms

5.Comment: In the conclusion section, it is advised to expand the discussion to encompass both the simulation and experimental results, providing a comprehensive overview of the findings.

  1. Reply:

Thanks for your valuable suggestion. We have modified the conclusions as follows

This paper proposes a brake-by-wire unit based on direct drive pump-valve cooperative, which has the advantages of rapid response and no deterioration of wheel side space and unsprung mass. The co-simulation model of the brake unit is established based on the Simulink-AMESim co-simulation. The pressure experiment shows that it has good pressure control performance, and the experimental curve is close to the simulation curve, which proves the accuracy of the simulation model. The feasibility of the simulation is further verified by ABS simulation, which provides a new implementation scheme for automotive distributed braking system. The characteristics of the key parameters are analyzed, and the influence of each parameter on the control performance is researched by changing the percentage of parameter change. This lays the foundation for further research. The structure optimization of electromagnetic linear actuator and the design of hydraulic control algorithm are the key to improve the performance of brake unit, and also the main research direction in the future.

Reviewer 3 Report

The paper proposes a new, more compact design of EHB unit that has advantages over previous works in this area. The physical design is presented and co-simulation and prototype hardware were utilized to validate the design and operation.

In the introduction section, the layout and workings of the new EHB are described and benefits of packaging are also provided.  I know that a production implementation image of the hardware is not possible at the time of writing the paper, but a concept style sketch or diagram of the physical proposed EHB hardware compared to conventional braking master cylinder or prior EHB systems might be beneficial.  Even taking images from prior EHB’s and indicating proposed changes or areas of opportunity might add significant value beyond current Figure 1’s hydraulic only schematic.

Figure 1 EMLA abbreviation is not defined in text prior to the fiure.

Glad to read that the authors utilized co-simulation of AMESim and Matlab/Simulink.  This is a powerful combination and AMESim’s hydraulic modeling library and multi-domain simulation capability is much more useful and appropriate than a model built in Simulink alone.  However, AMESim’s benefits in hydraulic representation are excellent, the simulation time for hydraulics and discontinuities can be a disadvantage. It would be good for the authors to describe any challenges there in terms of run time, discontinuities and in general the simulation to real time realized with the co-simulation setup.

At the end of section 3 it would be good to at least see a simulation, recommended would be the general maneuvers or tests that would be contained in Section 4 to give a preview of what the EHB is expected to do and the expectations for performance.

Results section is good, showing good correlation with simulation and leveraging the co-simulation environment to explore parameters to understand and improve performance for future hardware. Discussion length is adequate and well written.

The conclusion section is short, but concise for the material.  It would be good to include a few specific finds from section 4 that reinforce the benefits and claims in the introduction relative to previous EHB designs.

Author Response

Comments to the Author

The paper proposes a new, more compact design of EHB unit that has advantages over previous works in this area. The physical design is presented and co-simulation and prototype hardware were utilized to validate the design and operation.

Reply:

Thank you very much for your professional review of our article. We sincerely appreciate the concerns and questions you raised.

We take your suggestions very seriously and we have taken them seriously and extensively revised our previous manuscripts, and these changes have been important in improving the quality and accuracy of the article. Below are responses to each point of your comments.

1.Comment: In the introduction section, the layout and workings of the new EHB are described and benefits of packaging are also provided. I know that a production implementation image of the hardware is not possible at the time of writing the paper, but a concept style sketch or diagram of the physical proposed EHB hardware compared to conventional braking master cylinder or prior EHB systems might be beneficial. Even taking images from prior EHB’s and indicating proposed changes or areas of opportunity might add significant value beyond current Figure 1’s hydraulic only schematic.

  1. Reply:

Thank you for your valuable suggestion. In order to let readers have a more detailed understanding of the structure, a prototype is added in Figure 1, as follows

(a)

(b)

Figure 1. The structure of brake-by-wire unit based on direct drive pump-valve cooperative: (a) The schematic diagram; (b) the prototype.

2.Comment: Glad to read that the authors utilized co-simulation of AMESim and Matlab/Simulink. This is a powerful combination and AMESim’s hydraulic modeling library and multi-domain simulation capability is much more useful and appropriate than a model built in Simulink alone. However, AMESim’s benefits in hydraulic representation are excellent, the simulation time for hydraulics and discontinuities can be a disadvantage. It would be good for the authors to describe any challenges there in terms of run time, discontinuities and in general the simulation to real time realized with the co-simulation setup.

  1. Reply:

Thank you for your valuable suggestion. We set AMESim and Simulink as discrete simulation with fixed step length, and carry out data transmission and calculation of the two software in each step (5e-5s).

3.Comment: At the end of section 3 it would be good to at least see a simulation, recommended would be the general maneuvers or tests that would be contained in Section 4 to give a preview of what the EHB is expected to do and the expectations for performance.

  1. Reply:

Thank you for your valuable suggestions, which are of great help to the improvement of our paper. We need to conduct simulation verification after establishing the model. This part of the simulation content is combined with the experiment to section 4.

Reviewer 4 Report

The paper is not suitable for publication due to the following reasons:

1. While the paper describes a brake-by-wire unit design, it does not sufficiently emphasize the novelty or advancement of existing solutions in the field of automotive braking systems. It's important to clearly establish the unique contributions and potential impact of your work in addressing the challenges of distributed braking and automatic driving.

2. The paper briefly introduces the design concept but lacks crucial technical details that would allow readers to understand the architecture, components, and operational principles of the proposed brake-by-wire system. Without a comprehensive explanation of the design, it is challenging for the readers to assess its feasibility and potential benefits.

3. The paper briefly mentions the creation of an experimental platform and the verification of the co-simulation model through pressure control experiments and ABS simulations. However, the paper does not provide an in-depth analysis of the experimental results, including quantitative data and comparisons with expected outcomes. A thorough evaluation of the experimental results is necessary to validate the effectiveness and feasibility of the proposed system.

4. The paper lacks a comprehensive literature review that contextualizes the proposed brake-by-wire system within the broader landscape of brake-by-wire technologies, automatic driving, and distributed braking systems. Providing this context would help readers understand the significance and relevance of your work in the larger field.

5. The writing style and organization of the paper could be improved for better clarity. Sections should be organized in a logical order, and concepts should be explained in a clear and concise manner. Diagrams, figures, or visual aids could enhance the understanding of the proposed system.

6. The paper lacks a discussion on the potential challenges, limitations, and drawbacks of the proposed brake-by-wire system. It's important to address these aspects to provide a balanced perspective on the feasibility and practicality of the system in real-world scenarios.

Author Response

Comments to the Author

The paper is not suitable for publication due to the following reasons.

Reply:

Thank you very much for your professional review of our article. We sincerely appreciate the concerns and questions you raised.

We take your suggestions very seriously and we have taken them seriously and extensively revised our previous manuscripts, and these changes have been important in improving the quality and accuracy of the article. Below are responses to each point of your comments.

1.Comment: While the paper describes a brake-by-wire unit design, it does not sufficiently emphasize the novelty or advancement of existing solutions in the field of automotive braking systems. It's important to clearly establish the unique contributions and potential impact of your work in addressing the challenges of distributed braking and automatic driving.

  1. Reply:

Thanks for your valuable suggestions, the main contributions are as follows

The main contributions are summarized as follows. (1) A brake-by-wire unit based on direct drive pump-valve cooperative is proposed, which has the advantages of rapid response and no deterioration of wheel side space and unsprung mass. (2) Based on the Simulink-AMESim co-simulation platform, a co-simulation model of the brake unit is established and verified. (3) The characteristics of the key parameters are analyzed, and the influence law of each parameter on the control performance is investigated.

2.Comment: The paper briefly introduces the design concept but lacks crucial technical details that would allow readers to understand the architecture, components, and operational principles of the proposed brake-by-wire system. Without a comprehensive explanation of the design, it is challenging for the readers to assess its feasibility and potential benefits.

  1. Reply:

Thanks for your valuable suggestion, we have modified the description and structure diagram of the brake unit as follows:

This paper designed a brake-by-wire unit based on direct drive pump-valve cooperative. The structural mainly includes an electromagnetic linear actuator, hydraulic pump, active valve group, brake wheel cylinder based on the existing structure, and oil replenishment tank. The electromagnetic linear actuator, hydraulic pump, active valve group and oil replenishment tank are integrated and installed on the vehicle frame. The oil pipe is connected to brake wheel cylinder based on the existing structure, which effectively simplifies the brake pipe, and realizes distributed braking without deteriorating the wheel side space and unsprung mass. The electromagnetic linear actuator directly drives the reciprocating motion of the piston of the hydraulic pump, combined with the switch control of the active valve group, realize the rapid adjustment of the driving wheel cylinder pressure. The active valve group is composed of two solenoid valves, and the oil refill valve is a normally closed valve, which controls the continuity of the oil circuit between the oil replenishment tank and the direct drive pump, and the pressure maintaining valve is a normally open valve that controls the oil circuit from the pump to the brake wheel cylinder. The diameter of the piston in the direct drive pump is smaller than the diameter of the wheel cylinder piston, and the driving force of the actuator is amplified by the principle of unequal diameter hydraulic amplification, thus obtaining sufficient braking force using a smaller actuator. As the power source of the brake unit, the electromagnetic linear actuator completes the conversion of electrical energy to mechanical energy. The structure of brake-by-wire unit based on direct drive pump-valve cooperative is shown in Figure 1.

(a)

(b)

Figure 1. The structure of brake-by-wire unit based on direct drive pump-valve cooperative: (a) The schematic diagram; (b) the prototype.

3.Comment: The paper briefly mentions the creation of an experimental platform and the verification of the co-simulation model through pressure control experiments and ABS simulations. However, the paper does not provide an in-depth analysis of the experimental results, including quantitative data and comparisons with expected outcomes. A thorough evaluation of the experimental results is necessary to validate the effectiveness and feasibility of the proposed system.

  1. Reply:

Thanks for your valuable suggestion, we have revised the experimental results as follows

The experimental results of the electromagnetic linear actuator and active valves under the PID controller are shown in Figure 5. When the system reaches the target pressure and the working mode is switched to the pressure maintaining mode, the pressure maintaining valve closes, causing small fluctuations in the hydraulic pressure. At the same time, the actuator control signal is 0, and the current value rapidly decreases. Under the pressure in the pump chamber, the actuator moves towards the reset direction for a period of displacement.

To verify the hydraulic pressure tracking performance of the brake-by-wire unit based on direct drive pump-valve cooperative, the target sinusoidal pressure signal frequency is set to 2.5Hz and amplitude to 2.5MPa, and the hydraulic pressure tracking curve and the system state in the experiment are shown in Figure 6. PID control can track sinusoidal signal well. In the process of sine wave target tracking, the average error of simulation and experiment PID control is 0.197MPa and 0.218MPa and the amplitude attenuation of simulation and experiment PID control is 0.106MPa and 0.121MPa, respectively. The results show that the braking unit can effectively control the hydraulic pressure under sine wave target. However, due to the limitation of DC power supply voltage and sensor measurement accuracy, as well as the difficulty of system oil filling in actual conditions, the actual pressure curve differs from the simulation.

To further simulate the pressure tracking performance, set the target signal of derivative mutation, the target hydraulic pressure frequency of the triangular wave braking is set to 2.5Hz and the amplitude to 2.5MPa. Hydraulic pressure tracking curve and the system state in the experiment are shown in Figure 7. At 0.2s and 0.4s, the derivative of the target signal suddenly changes, but the PID controller can still track triangular wave signal well. Due to the hydraulic oil in the system not being fully filled, the actuator position at 0.4s is higher than the initial state. In the process of triangular wave target tracking, the average error of simulation and experiment PID control is 0.141 and 0.203, respectively. The results show that the proposed brake unit has good accuracy when maintaining a certain rate of increasing or decreasing hydraulic pressure, which verifies its good tracking performance.

4.Comment: The paper lacks a comprehensive literature review that contextualizes the proposed brake-by-wire system within the broader landscape of brake-by-wire technologies, automatic driving, and distributed braking systems. Providing this context would help readers understand the significance and relevance of your work in the larger field.

  1. Reply:

Thank you for your valuable suggestion, we add an overview of the distributed braking system in the introduction, and propose the advantages of the braking unit designed in this paper. The specific introduction is modified as follows

With the development of vehicle chassis towards the direction of X-by-wire chassis and slide chassis, the demand for distributed braking system is becoming more and more intense. At the same time, high-level automatic driving puts forward higher requirements for the response speed and control accuracy of the drive-by-wire system [1-4].

Electronic Hydraulic Brake (EHB) System has been widely favored due to its advantages such as rapid response, high power density and brake structure compatibility [5-9]. In 2013, Bosch launched I-Booster, a motor servo booster that is independent of vacuum booster, which is a typical motor servo EHB system. TRW introduced a high-pressure accumulator-based EHB system called the SCB, which has a brake master cylinder consisting of front and rear chambers and parallel pistons. Since the pistons can be moved fore and aft and the front part is connected to the front wheel, the pressure of the system can be adjusted through the rear part of the pistons. [10]. In 2010, Hitachi launched the EHB system named e-ACT, which uses an electric motor to drive an actuator that pushes a master cylinder piston, and the rotational force of the motor is converted into linear motion by a ball screw. [11]. In 2021, Continental launched a new generation of EHB system MKC2 based on MKC1, which uses a multi-logic framework with independent partitions, successfully reducing the number of brake system components, and its modular and scalable design provides ideas for future dynamic control of brake systems.

At present, EHB system is still the mainstream of the development of brake-by-wire system. In order to take into account the needs of market and efficiency, it is of great significance to explore a new brake-by-wire structure to improve the performance of brake-by-wire and simplify the system structure [12-15]. Sun designed and developed an integrated master cylinder and decoupling electro-hydraulic composite braking system, and proposed the hydraulic braking force distribution strategy [16]. Wu designed an integrated EHB system. By controlling the solenoid valve and designing the working mode switching control strategy, the working mode switching was realized to achieve high redundancy and independent control of four wheel cylinders of the braking system [17]. EHB with different characteristics have been proposed to promote the development of brake-by-wire technology, but it is difficult to directly apply to distributed braking systems [18-20]. It is necessary to design a distributed brake-by-wire unit to meet the needs of intelligent driving, and meet the requirements of new energy vehicles for precise control of braking force and lightweight. Fu designed a new type of electromechanical brake with automatic wear adjustment function, and adopted worm gear mechanism for power transmission, which has the advantages of low delay and small size [21]. Yu designed a distributed electro-hydraulic brake-by-wire system, which uses a motion conversion mechanism to change the motion of the rotating motor into linear motion and then drive the hydraulic wheel cylinder [22]. Chang designed an EHB unit that uses a linear motor to drive the hydraulic piston, amplifying the driving force of the motor through unequal diameter hydraulic pistons and wheel cylinder pistons [23]. The existing EHB are often improved based on proven hydraulic braking systems, which still retain complex hydraulic pipelines [24-26]. Distributed brake-by-wire systems often require the entire brake unit to be integrated into the caliper, deteriorating wheel side space and unsprung mass. Therefore, This paper designed a brake-by-wire unit based on direct drive pump-valve cooperative, through the electromagnetic linear actuator directly driven by the hydraulic pump and the active valve coordination to achieve wheel cylinder pressure regulation.

The main contributions are summarized as follows. (1) A brake-by-wire unit based on direct drive pump-valve cooperative is proposed, which has the advantages of rapid response and no deterioration of wheel side space and unsprung mass. (2) Based on the Simulink-AMESim co-simulation platform, a co-simulation model of the brake unit is established and verified. (3) The characteristics of the key parameters are analyzed, and the influence law of each parameter on the control performance is investigated. The rest of this work is organized as follows. Parameter matching design of the key parameters is carried out in Section 2. The dynamic model of the brake-by-wire unit based on direct drive pump-valve cooperative is established in Section 3. The feasibility is verified by pressure experiments and ABS simulations in Section 4. Finally, conclusion is made in Section 5.

5.Comment: The writing style and organization of the paper could be improved for better clarity. Sections should be organized in a logical order, and concepts should be explained in a clear and concise manner. Diagrams, figures, or visual aids could enhance the understanding of the proposed system.

  1. Reply:

Thanks for your valuable suggestion. The writing style and organization of the paper has been revised, marked in blue font, and the specific reorganized chapters are as follows:

  1. Introduction
  2. System scheme and principle
  3. Parameter matching design
  4. System modeling

3.1. Modeling of the electromagnetic linear actuator

3.2. Hydraulic system modeling

3.3. quarter-car dynamics model establishment

3.4. Co-simulation model construction

  1. Results and analysis

4.1. The experimental platform

4.2. Analysis of hydraulic pressure response performance

4.3. Analysis of Hydraulic Pressure Tracking Performance

4.4. Analysis of the impact of parameters on system performance

4.4.1. Actuator parameter

4.4.2. Hydraulic system parameter

4.5. ABS performance analysis

  1. Conclusions

6.Comment: The paper lacks a discussion on the potential challenges, limitations, and drawbacks of the proposed brake-by-wire system. It's important to address these aspects to provide a balanced perspective on the feasibility and practicality of the system in real-world scenarios.

  1. Reply:

Thanks for your valuable suggestion. This paper proposes a brake-by-wire unit based on direct drive pump-valve cooperative, which has the advantages of rapid response and no deterioration of wheel side space and unsprung mass. At present, the limitation of the brake unit is that the actuator with higher power and higher accuracy is needed to boost the brake wheel cylinder. The structure optimization of electromagnetic linear actuator and the design of control algorithm are the key to improve the performance of brake unit, and also the main research direction in the future. The prototype has been verified by experiment, and the result shows that the braking unit has good dynamic response and steady-state tracking effect, which has reference significance for the practicability.

Round 2

Reviewer 1 Report

The author made all of the recommended changes, and the paper can now be accepted.

Reviewer 4 Report

The paper is revised carefully and now it can be accepted in its present form.